# ON ANYTIME LEARNING AT MACROSCALE

## ABSTRACT

Classical machine learning frameworks assume access to a possibly large dataset in order to train a predictive model. In many practical applications however, data does not arrive all at once, but in large batches over time. This creates a natural trade-off between accuracy of a model and time to obtain such a model. A greedy predictor could produce non-trivial predictions by immediately training on batches as soon as these become available but, it may also make sub-optimal use of future data. On the other hand, a tardy predictor could wait for a long time to aggregate several batches into a larger dataset, but ultimately deliver a much better performance. In this work, we consider such a streaming learning setting, which we dub *anytime learning at macroscale* (ALMA). It is an instance of anytime learning applied not at the level of a single chunk of data, but at the level of the entire sequence of large batches. We first formalize this learning setting, we then introduce metrics to assess how well learners perform on the given task for a given memory and compute budget, and finally we test about thirty baseline approaches on three standard benchmarks repurposed for anytime learning at macroscale. Our findings indicate that no model strikes the best trade-off across the board. While replay-based methods attain the lowest error rate, they also incur in a 5 to 10 times increase of compute. Approaches that grow capacity over time do offer better scaling in terms of training flops, but they also underperform simpler ensembling methods in terms of error rate. Overall, ALMA offers both a good abstraction of the typical learning setting faced everyday by practitioners, and a set of unsolved modeling problems for those interested in efficient learning of dynamic models.

## 1 INTRODUCTION

Empirical risk minimization (Vapnik, 1998) is the dominant framework to formalize the learning process of a supervised task, and it has been critical to the success of large scale training of deep learning systems on a wide variety of applications. Within this framework, training data is assumed to be provided to the learner all at once. Alternatively, when the dataset is very large (essentially infinite), data is streamed to the learner one minibatch at the time, assuming that the rate at which samples are received matches the model's processing time to learn from them.

Learning over streams of data has been studied in the machine learning domain for a long time (see Section 2 and Figure 1 for more details) with different assumptions: for instance in online learning, it is usually assumed that datapoints are coming one by one and have to be processed as soon as they are received. In continual learning, the streaming of data usually corresponds to a stream of large datasets corresponding to different tasks to solve, etc. In this paper, we define a simple yet important setting where there is a single task to solve, and where training data often comes at a *slower* rate than a model can process it. Moreover, it comes in relatively large batches once in a while. While poorly studied, this setting corresponds to practical applications encountered in production pipelines. For instance, it is faced by teams deploying language modeling applications (e.g content moderation) build models that are trained on large amounts of data like filtered versions of Common Crawl, which are dumps of the internet. However, new snapshots are available every month, as new content is generated over time. Therefore datasets keep getting bigger every few months and models need to be retrained accordingly. Similarly, visual object recognition datasets used in deployed applications are often extended every few months to include new images with their corresponding annotations.

---

* Authors contributed equally

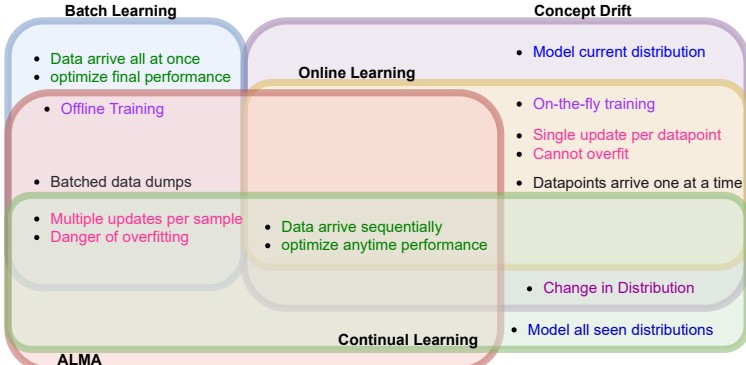

Figure 1: ALMA compared to other learning frameworks.

Practically, there are two main approaches to integrate information present in a new batch of data in an existing model. If a lot of computational resources are available, a new and bigger model is instantiated and trained *from scratch* on the union of the old training set with the new batch of data. However, since this is a computationally very intensive process, retraining is typically done only rarely, once several batches of data have been collected. We call this approach "tardy" large-scale learning, since a predictor is available only at a later time. Another option, particularly suitable when computational resources are scarce and a predictor is needed quickly, is to simply finetune the old model on the new data as this arrives. Note that, in that settings, methods from the data stream domain or from the online learning domain that are based on the idea of processing any datapoint just once are not suitable since they have been developed for different use-cases.

This trade-off is emblematic of *anytime learning*, a learning setting where a learner has to provide good predictions at any point in time, while improving its performance over time as more and more data is observed. From an anytime learning perspective, neither training a large model after all data is received nor finetuning on the newly added batch of data are not satisfying. The former approach is a poor anytime learner because one needs to wait for a long time before obtaining a useful predictor. The latter approach is a poor anytime learner because it typically cannot leverage very well future batches of data since the model has a fixed capacity, determined on a small portion of the overall dataset and because inherently the model is trained on non i.i.d. data.

In this work, we aim at exploring this accuracy versus time trade-off of anytime learning, not at the level of a single batch of data, but at the macroscale of the entire sequence of batches. This is a setting which more closely mimics practical applications, that we call *anytime learning at mascroscale* (ALMA). In this learning setting, we assume that the time to train a model is negligible compared to the interval of time between two consecutive batches of data (and therefore we do not care about how quickly a learner adapts to a new batch), yet efficiency matters in the sense that for the same performance a predictor that uses less compute and memory is preferable. In summary, we are interested in a learner that i) yields high accuracy, ii) can make non-trivial predictions at any point in time while iii) limiting its computational and memory resources.

Our first contribution is to formalize the ALMA problem and to introduce metrics to evaluate learners (§3). We consider three different axes: error rate, memory and amount of computation. By measuring these quantities against time, via an area under the curve, we account not only for the final performance but also for the whole training trajectory over the sequence of large batches of data.

Our second contribution is an extensive empirical evaluation (§5) of various models (§4) that strike different trade-offs between accuracy and time to obtain a useful predictor. In particular, we explore models that fall in between greedy finetuning and tardy large-scale learning, and investigate models that leverage batches of data at an intermediate rate. We also consider a rich family of modular architectures, from plain ensembling methods to hierarchical mixture of experts, and several variants thereof, including those that have access to a replay buffer storing all previous batches of data and those that can grow capacity over time.

Our findings across three different benchmarks, including a large scale language modeling one, can be summarized as follows. a) An intermediate waiting time offers the best trade-off between accuracy and time to yield such a predictor. However, b) there is no single approach striking the best trade-off between performance and efficiency for various model sizes. c) Retraining from scratch a big model does offer the lowest error rate but sacrifices efficiency. d) Interestingly, large models are the most statistically efficient even when considering small datasets (like MNIST) and fully

connected networks. e) While approaches to grow capacity exhibit gains in terms of computational efficiency, these do not even outpeform simple ensembles. Overall, our work points at several research opportunities to improve modeling in a streaming setting of broad practical relevance, rather then pointing at any particular solution. We have also released code to reproduce our experiments and the entire platform implementing ALMA.

## 2 RELATED WORK

ALMA relates to several other learning frameworks: offline learning, continual learning, online learning and transfer learning as illustrated in Figure 1. i) It shares the same assumptions of classical empirical risk minimization (ERM) (Vapnik, 1998) at the level of each batch of data. However, it overall violates ERM's assumptions of i.i.d. observations, because data points come in a stream of data chunks. ii) Because of this, ALMA relates to continual learning (CL) (Ring, 1994; Thrun, 1994; Ring, 1997; Thrun, 1998), with the key difference that the data distribution across batches (or tasks) is assumed stationary in ALMA. Therefore, ALMA can be seen as a special case of CL with a single task to solve. iii) ALMA relates also to online learning (Bottou, 1998) since it assumes that data are coming in a stream, an assumption also made in the concept drift literature (Lu et al., 2018). However, in online learning examples are streamed one at the time (or at random from a large dataset), while in ALMA the learner receives large batches of data sequentially In ALMA, received data can be processed multiple times as opposite to the online learning setting that usually assumes that any new datapoint has to be processed as soon as it is available, and will not be reused in future updates. iv) Finally, ALMA relates more broadly to transfer learning (Pan & Yang, 2010), as the problem of adapting to a new batch of data can be interpreted as leveraging knowledge acquired on previous batches to more effciently learn from the new batch of data.

Of course, ALMA relates to anytime learning (Grefenstette & Ramsey, 1992; Ramsey & Grefenstette, 1994), which has been recently applied to compare various autoML frameworks (Liu et al., 2020). However, in this work we are not interested in assessing the anytime learning ability at the level of each chunk of data, but only at a coarser granularity, at the level of the entire stream of chunks. Inspired by Liu et al. (2020), we consider the area under the curve of error rate against time to measure performance, but in order to account also for compute and memory budget, we add to our evaluation metrics also the area under the curve for memory and compute.

From the more theoretical side, there has been work about sub-bagging (Bühlmann & Yu, 2002) (bagging using subsets of a larger dataset) which is similar to our setting but without the sequential aspect of it. In this context, Breiman (1999) proposed a model similar to our growing ensembling (gEns), Bühlmann & Yu (2002) studied sub-bagging as a way to make the prediction of tree classifiers more robust while Zou et al. (2021) studied the consistency of the estimator in this setting. We defer to future studies the analysis of ALMA, while in this work we focus on the empirical evaluation.

Shifting the discussion to prior work on models that adjust their capacity dynamically, Waterhouse & Robinson (1995) introduced an approach to grow a hierarchical mixture of experts model (Jordan & Jacobs, 1994). This is a tree structured model where experts are at the leaves and gating functions are at non-terminal nodes. The tree determines a hierarchical partition of the input space into regions that are associated to each expert. This approach was made more efficient in later work by (Fritsch et al., 1996). In this work we consider a baseline (gMoE) that extends this prior work to hierarchical mixture of experts (Eigen et al., 2014; Denoyer & Gallinari, 2015; Lepikhin et al., 2020).

Growing architectures have also been studied in CL. For instance, Fernando et al. (2017) and Veniat et al. (2021) proposed a modular architecture that is assembled for every task, possibly reusing previously trained modules. The major difference with our work is that in our case routing is input dependent as opposed to task dependent. Yoon et al. (2018) instead proposed a method to incrementally and smoothly add hidden units. Similarly, Wen et al. (2020) proposed a heuristic approach to automatically adjust the network depth. Wang et al. (2017) considered growing both depth and width when finetuning to a new task. Liu et al. (2019a) and Wu et al. (2020) proposed approaches to grow architectures in depth and width by leveraging Taylor approximation and greedy selection. In our work, we benchmark against this last variant. None of these approaches have been applied to the ALMA setting to date.

Finally, some of our findings are built upon and extend recent empirical evaluations studying the scaling properties of language models (Kaplan et al., 2020a; Li et al., 2020b). In this study, we

confirm the conclusion that bigger models generalize better and are more statistically efficient, not only in language modeling tasks using a transformer architecture, but also in smaller scale computer vision tasks using both fully connected and convolutional architectures.

## 3 LEARNING SETTING

In anytime learning at macroscale (ALMA), we assume that there exists an underlying data distribution $p(x, y)$ with input $x \in \mathbb{R}^D$ and desired label $y \in \{1, \ldots, C\}$. Notice that extensions to regression and unsupervised learning (where $y$ is missing) are trivial, and therefore in this work we focus on classification problems for simplicity of exposition. A important property of ALMA is that data is presented to the learner as a stream $\mathcal{S}_B$ of $B$ consecutive batches of examples. Let $\mathcal{D}_i$ be a collection of $N \gg 0$ i.i.d. samples randomly drawn from $p(x, y)$, for $i \in \{1, \ldots, B\}$. The stream is then defined as the ordered sequence $\mathcal{S}_B = \{\mathcal{D}_1, \ldots, \mathcal{D}_B\}$. We refer to each dataset $\mathcal{D}_i$ as *mega-batch*, as it is composed by a large number of examples. Typically a learner $m : \mathbb{R}^D \to \{1, \ldots, C\}$ updates its parameters by processing a *mini-batch* of $n \ll N$ examples at the time from each mega-batch $\mathcal{D}_i$, and by iterating several times over each mega-batch before being presented with the next mega-batch. Since the learner cannot access future mega-batches, overall the data distribution is not i.i.d., even though samples drawn from each mega-batch are i.i.d., and cross-validation is performed using a subset of the current mega-batch. A learner could decide to use previous mega-batches when learning on the current mega-batch, but this will increase its compute usage.

Finally, we assume that the time it takes a learner to update its internal parameters after having observed a mega-batch is much less than the interval between the arrival of two consecutive mega-batches. In other words, the rate at which data arrives is slower than the processing time of the model, and therefore the model could decide to iterate several times over the data at its disposal to improve its prediction accuracy.

### 3.1 METRICS

We evaluate learners in the ALMA setting across three axes, namely: accuracy, memory and computation. Let $t$ be the time at which the $t$-th mega-batch arrives; this data can be used by the model to update its parameters or it is simply aggregated to previous mega-batches for later use.

We compute the error rate of model $m$ at time $t$ (after the arrival of the $t$-th mega-batch) and compute the area under the curve obtained varying $t$ from 0 till the total number of mega-batches $B$; the resulting cumulative error rate (CER) is:

$$\text{CER} = \sum_{t=0}^{B} \frac{1}{|\mathcal{D}^{\text{TS}}|} \sum_{(x,y) \in \mathcal{D}^{\text{TS}}} |m(x; \theta_t) \neq y| \tag{1}$$

where $m(x; \theta_t)$ is the model at time $t$ equipped with parameters $\theta_t$, $\mathcal{D}^{\text{TS}}$ is the test set, $|\mathcal{D}^{\text{TS}}|$ is the number of examples in the test set, and $|m(x; \theta_t) \neq y|$ is one if the model prediction does not match the ground truth label and zero otherwise. The outer sum computes the discrete integral of the error rate over time. CER is going to be small only when the error rate is small throughout the whole stream . CER is instead large for a tardy model that waits till the very last mega-batch to update the model, even though eventually this may obtain a very low final error rate. If not perfect, CER provides a good summary of the performance of a system across time. Anyway, to fully capture the differences between two models, it is needed to have a deeper look at the performance across time as illustrated in Figure 2 for instance.

Similarly, we compute the cumulative memory usage and compute as:

$$\text{Mem} = \sum_{t=0}^{B} |\theta_t|, \quad \text{Comp} = \sum_{t=0}^{B} \mathcal{O}(m(\cdot; \theta_t)) \tag{2}$$

where $|\theta_t|$ is the number of free parameters of the model at time $t$, and $\mathcal{O}(m(\cdot; \theta_t))$ is the number of flops used by the model to process the $t$-th mega-batch. Once again, by measuring the area under the curves obtained by tracking these quantities over time we obtain a holistic assessment of memory and compute throughout the whole stream. A model can obtain small Mem and Comp only if it does not consume memory and if it is computationally parsimonious throughout the entire duration of the stream.

---

**Algorithm 1** Training in the ALMA setting

---

 1: **procedure** TRAIN($m, w$, replay, grow)                      ▷ $m$ is the model, $w$ is the waiting time
 2:      $t \leftarrow 1$
 3:      $\mathcal{D} \leftarrow \emptyset$
 4:      **while** $t < B$ **do**                                                           ▷ For each stage
 5:          **if** replay **then**                                                ▷ Acquire $w$ mega-batches
 6:              $\mathcal{D} \leftarrow \mathcal{D} \cup \mathcal{D}_t \cup ... \cup \mathcal{D}_{t+w-1}$
 7:          **else**
 8:              $\mathcal{D} \leftarrow \mathcal{D}_t \cup ... \cup \mathcal{D}_{t+w-1}$
 9:          $t \leftarrow t + w$
10:          **if** grow **then**
11:              $m.grow()$                            ▷ Grow the model if the model is a growing model
12:          $m.train(\mathcal{D})$               ▷ Fine-tune or retrain from scratch $m$ on the collected dataset

---

## 4 LEARNING ALGORITHMS

In this section, we describe the methods we tested in the ALMA setting. They generally follow the learning procedure shown in Algorithm 1. At a high level, we consider two families of models, those with a monolithic architecture and those with a modular architecture (e.g., ensembling). The latter are amenable to grow over time by adding new modules to the existing set. We will start by describing fixed architectures (§4.1) and then conclude with growing architectures (§4.2). All models are also given the option to replay previous mega-batches.

### 4.1 FIXED ARCHITECTURES

The first family of methods trains models with a fixed architecture. These models are sequentially trained over new mega-batches and exhibit a fixed memory footprint. We consider three models:

**Single Model (*SM*):**    This is a standard multi-layer neural network (e.g., fully connected neural network or transformer) trained by stochastic gradient descent. It can be initialized from random or from the parameters of the model trained on the previous mega-batch. The initializaiton choice is determined via cross-validation.

**Ensemble of Models (*Ens*):**    The second approach is the simplest modular approach, consisting of an ensemble of $N$ neural networks with the same architecture, each being trained independently on the same sequence of data. The output of the overall model at test time is the average probability distribution produced by each component[1]. The advantage of *Ens* is that training and inference can be trivially parallelized, enabling to scale up model parameters very easily. The disadvantage is that inference requires $N$ times more compute than what is required by each component.

**Uniform Mixture of Models (*UMix*):**    A potential drawback of *Ens* is that evaluation and training are inconsistent. *UMix* addresses this by training a model whose prediction is the average (in logit space) of the predictions produced by $N$ networks. While this requires synchronization during training, now both training and evaluation use the same model.

### 4.2 GROWING ARCHITECTURES

In the previous section, the number of parameters and the architecture of the model are fixed throughout the model's lifetime. However, as more data is observed, it is interesting to consider dynamic architectures that grow over time, because these may save compute and memory during the earlier stages of learning while providing more predictive power during the later stages. We consider three growing approaches:

---

[1]Classical bagging approaches and majority vote strategies have been also explored without significant difference.

**Growing Ensemble (*gEns*):**    Like the *Ens* model, *gEns* is also a combination of neural networks trained independently. While *Ens* considers $N$ networks that are, at each stage, trained over the new chunck of data, *gEns* replaces this step by a growing step where $n$ neural networks are added. In our implementation, only these $n$ neural networks are trained over the new data, while the other neural networks (trained on previous mega-batches) are kept fixed.

**Growing Mixture of Experts (*gMoE*):**    A hierarchical mixture of experts models (MoE) is an architecture where at layer $l$ the output representation $z^l$ is: $z^l = \sum_{j=1}^{k} g(j|z^{l-1})h(z^{l-1}|j)$, where $g$ is the gating or routing function and $h(\cdot|j)$ is the $j$-th expert. Compared to *Ens*, MoE has exponentially many more components albeit with a lot of parameter sharing. Another advantage is that by selecting only one (or a few) experts, the computational cost is independent of the number of experts, assuming the cost of gating is negligible compared to the cost or executing the experts. The main issue is that MoE are notoriously harder to train (Eigen et al., 2014; Denoyer & Gallinari, 2015; Lepikhin et al., 2020). In this work, we consider a growing version of MoE, which we denote with *gMoE*, whereby experts are added over time. See Appendix A for more details.

**Firefly (Wu et al., 2020) (*FF*):**    *FF* is a method which progressively grows neural networks, jointly optimizing both the model architecture and parameters. Growth includes both a width expansion by adding new hidden units (or feature maps) as well as a depth expansion by adding new layers. Importantly, this is an example of non-modular method unlike *Ens* or *gMoE*, which is potentially more expressive but also more inefficient at inference time because there is no structured sparsity that can be leveraged to speed up computation.

## 5    EXPERIMENTS

In this section we first describe how standard benchmarks can be repurposed for ALMA, we then provide the details of the models we tested, and we finally conclude with an analysis of the results we obtained, aiming to understand which method attains the best trade-off between time, accuracy, compute and memory usage.

**Datasets**    We consider a variety of datasets. The first dataset is MNIST (LeCun et al., 1998), which consists of a training set with 60,000 quasi-binary handwritten digits of size 28x28 pixels, and a test set with 10,000 examples. The second dataset is CIFAR 10 (Krizhevsky, 2009) that has a training set with 50,000 images of size 32x32 pixels belonging to 10 classes such as bird, car, horse, ship, truck, etc. The third dataset, used for our large-scale language modeling evaluation, is a portion of the collection of English language text introduced in Liu et al. (2019b), consisting of Books, Wikipedia and Common Crawl. We consider 4 (large) mega-batches for training and one additional mega-batch for evaluation, each consisting of approximately 440M words; we also hold out a validation set with approximately 0.5M words of Common Crawl for model selection. We use a byte-pair encoding (BPE) (Sennrich et al., 2016) vocabulary with 50,000 units, following Radford et al. (2019). This dataset is fairly representative of what practitioners might face when maintaining a deployed system with new data arriving every few months.

Given a dataset like any of the above, we construct a benchmark for ALMA evaluation as follows: 1) we randomly partition the training set into $B$ mega-batches with equal number of training examples ($B = 50$ for MNIST and CIFAR 10, and 4 for the text dataset), 2) from each mega-batch we extract 10% of the data to build the mega-batch validation set (except for the large scale language modeling dataset where we use the provided validation set), and 3) we create a learning experience by doing one pass over the sequence of mega-batches. For each mega-batch, the learner can query as many mini-batches as desired. The learner can also decide not to train on the data of a mega-batch right away but instead to wait and accumulate data across a few consecutive mega-batches. While the learner observes data, it is also tested on the test set. This is not used for validation purposes, but only for final reporting as shown in §5.1.

**Models**    We evaluate the six approaches presented in §4, and for each of them we consider various waiting times, a version with and without replay, and at least two model sizes. For each setting, we cross validate over several hyper-parameters such as initializaiton type, learning rate, stopping criterion, growth rate, etc.

Figure 2: Accuracy over time for small models (left) and large models (right) on CIFAR 10 with no replay.

| Method | $\|\theta\|$ | Cum. Train TFLOPS | Inference MFLOPS | CER |
|---|---|---|---|---|
| *SM* | 3210 | 0.02 (0.09) | 0.003 | $1054_{\pm25}$ ($1062_{\pm61}$) |
| | 16330 | 0.09 (0.48) | 0.016 | $631_{\pm10}$ ($616_{\pm9}$) |
| | 26506 | 0.14 (0.79) | 0.026 | $582_{\pm12}$ ($554_{\pm5}$) |
| | 132127 | 0.71 (3.92) | 0.132 | $504_{\pm8}$ ($485_{\pm4}$) |
| | 269322 | 1.45 (7.98) | 0.269 | $497_{\pm8}$ ($474_{\pm4}$) |
| *Ens* | 16050 | 0.09 (0.47) | 0.080 | $870_{\pm9}$ ($885_{\pm18}$) |
| | 132530 | 0.71 (3.93) | 0.661 | $517_{\pm6}$ ($493_{\pm5}$) |
| *UMix* | 16050 | 0.09 (0.47) | 0.080 | $668_{\pm17}$ ($646_{\pm18}$) |
| | 132530 | 0.71 (3.93) | 0.661 | $508_{\pm10}$ ($486_{\pm5}$) |
| *gEns* | $3210 \rightarrow 32100$ | 0.02 (0.09) | $0.003 \rightarrow 0.032$ | $1020_{\pm39}$ ($1027_{\pm29}$) |
| | $26506 \rightarrow 265060$ | 0.14 (0.79) | $0.026 \rightarrow 0.264$ | $551_{\pm6}$ ($543_{\pm7}$) |
| *gMoE* | $7950 \rightarrow 50610$ | 0.16 (1.08) | $0.008 \rightarrow 0.050$ | $863_{\pm17}$ ($766_{\pm16}$) |
| | $54318 \rightarrow 304626$ | 0.97 (6.55) | $0.054 \rightarrow 0.304$ | $573_{\pm8}$ ($535_{\pm5}$) |

Table 1: MNIST results with waiting time equal to 5. Rows with parameter size in blue refer to small networks (e.g., *Ens* totalling 16050 parameters has five components of the same size of *SM* shown in blue); similarly, rows in orange refer to the large network size. Numbers in parentheses are for versions with replay. For all metrics smaller is better.

Next, we describe in details the architecture used on each dataset. Further experimental details to aide reproducibility are reported in Appendix B. On MNIST the backbone architecture of *SM* is a three layer fully connected neural network with ReLU units. We considered two hidden units sizes, namely 4 and 32 (denoted by [s] and [b], respectively), which let us simulate the regime of big data relative to the size of the network and explore how to grow architectures without worrying about overfitting. Similarly, the components of *Ens*, *gEns* and *UMix* are *SM* networks of the same size as stated above; *gMoE* also starts off as *SM* and adds modules (at the first two layers) that have the same size as the original layer of *SM*. When varying the waiting time, i.e., the number of mega-batches that are aggregated before initiating a new training session, we use the suffix "_w" to indicate its value.

On CIFAR 10, the methods and notations are the same as in MNIST. The only difference is that the backbone architecture is a scaled down version of a VGG19 convolutional neural network (Simonyan & Zisserman, 2015), where the number of intermediate feature maps is the same for each layer and equal to either 4 or 32. On this dataset, we also consider *FF* starting off from the same VGG19 backbone.

For the language modeling task *SM* is a Switch Transformer (Fedus et al., 2021), which is a hard mixture of experts model with an additional load balancing loss term and hard capacity constraint applied during training to prevent uneven expert utilization. Following Fedus et al. (2021), we fix the weight of the balancing loss term to 0.01 and use a capacity factor of 1, ensuring relatively uniform expert utilization. We train the model using Adam (Kingma & Ba, 2015) and tune the learning rate and dropout on the validation set. In the growing setting we copy the expert weights and gating network weights corresponding to the top-$k$ experts incurring the largest loss, where $k$ is typically between 2 and 4. We consider two model sizes: a *base* model with 6 layers and model dimension of 512, for a total of 40M shared parameters and 6M additional parameters per expert; and a *large* model with 12 layers and model dimension of 768, for a total of 96M shared parameters and 28M additional parameters per expert. We use an input sequence length of 512 tokens and we do not use replay given the large chunk sizes.

## 5.1 RESULTS

In Fig. 2 we start by analyzing learning curves on CIFAR 10 for a subset of the methods as a function of the waiting time. We then dive into analyzing all methods on both MNIST (Tab. 1) and CIFAR 10

| Method | $|\theta|$ | Cum. Train TFLOPS | Inference MFLOPS | CER |
|---|---|---|---|---|
| SM | 2510 | 3.6 (20.0) | 0.4 | $2706_{\pm132}$ ($2643_{\pm39}$) |
| | 11710 | 15.0 (82.5) | 1.7 | $2038_{\pm32}$ ($1842_{\pm51}$) |
| | 31660 | 38.6 (212.5) | 4.3 | $1764_{\pm43}$ ($1561_{\pm7}$) |
| | 140970 | 164.9 (907.0) | 18.3 | $1524_{\pm23}$ ($1305_{\pm15}$) |
| | 1358510 | 1546.4 (8505.3) | 171.8 | $1307_{\pm19}$ ($1118_{\pm17}$) |
| Ens | 12550 | 18.2 (99.8) | 10.1 | $2440_{\pm29}$ ($2311_{\pm26}$) |
| | 704850 | 824.5 (4534.9) | 458.1 | $1230_{\pm10}$ ($1046_{\pm10}$) |
| UMix | 12550 | 18.2 (99.8) | 10.1 | $2087_{\pm36}$ ($1840_{\pm35}$) |
| | 704850 | 824.5 (4534.9) | 458.1 | $1502_{\pm36}$ ($1286_{\pm28}$) |
| gEns | 2510 → 25100 | 3.6 (20.0) | 0.4 → 4.0 | $2727_{\pm70}$ ($2542_{\pm24}$) |
| | 140970 → 1409700 | 164.9 (907.0) | 18.3 → 183.2 | $1348_{\pm17}$ ($1283_{\pm8}$) |
| gMoE | 5214 → 29550 | 10.1 (65.3) | 0.5 → 1.7 | $2401_{\pm61}$ ($2089_{\pm28}$) |
| | 27544 → 1485690 | 555.2 (3638.9) | 26.2 → 97.2 | $1448_{\pm29}$ ($1222_{\pm18}$) |
| FF | 2446 → 32519 | 26.1 (222.3) | 0.4 → 5.5 | $2272_{\pm26}$ ($1947_{\pm44}$) |
| | 140458 → 1646809 | 646.3 (7166.3) | 18.3 → 138.9 | $1450_{\pm36}$ ($1189_{\pm20}$) |

Table 2: CIFAR 10 results with waiting time equal to 5. Same notations and colors as in Table 1.

(Tab. 2), using the optimal empirical value of waiting time. We conclude by confirming the major findings at scale on the language modeling task (Tab. 3).

Fig. 2 shows the test error rate as a function of the number of mega-batches received for both the small (left) and the large (right) model. We observe that an intermediate waiting time (in this case equal to 5) strikes the best trade-off between accuracy and time for all methods, since curves with waiting time equal to 5 have the lowest area under the curve. Greedy methods using waiting time equal to 1 achieve lower error rate only during the very beginning of the stream. Second, we observe that bigger models (*SM* and *Ens*) not only generalize better but they are also statistically more efficient: the small *Ens* obtained almost 35% error rate by the end of its learning experience, which is worse than the error rate obtained by the large *Ens* just after having observed one tenth of the entire stream. The statistical efficiency of large models does not apply only to large transformers (Kaplan et al., 2020a), but also to fully connected (we obtained similar results on MNIST) and convolutional models.

Next, using the waiting time that yielded the lowest cumulative error rate, we compare all methods discussed in §4, focusing our discussion on Tab. 2 of CIFAR 10 as same conclusions apply to MNIST as well (see Tab. 1).

First, replay lowers the CER by a relative amount of about 10% at the cost of increasing the cumulative training flops by a factor of more than 5, which is rather substantial. Notice that retraining from scratch using memory replay, as reported here in parentheses, is nowadays the dominant approach to deal with sequential datasets.

Second, *Ens* works better than *UMix* for larger models, and vice versa. We surmise that ensembling may alleviate overfitting of large models, but coordinating the components of the ensemble like *UMix* does, is more effective in an underfitting regime (i.e with small models). *Ens* thus looks like a good method to train large architectures without suffering of the overfitting aspect and may be used when the complexity of the task is not known *a priori*.

Third, all growing approaches perform rather similarly, particularly when starting from larger back-bones, although they strike slightly different trade-offs. For instance, *gMoE* is the most efficient at test time, while *FF* yields a lower error rate. Interestingly, none of the approaches that grow architectures currently manages to beat *Ens* in terms of error rate when starting from a large backbone, although they require substantially fewer flops at inference time. Finally, while methods derived from *SM* (for the same size of the initial backbone, see rows with the same color in the table) all manage to beat *SM*, it is also worth noting that for the same number of parameters *SM* is still the best performing method, unless there is overfitting. In particular, *Ens* with 12550 parameters achieves a CER of 2440 while *SM* with 11710 parameters obtains a CER of 2038 while requiring much less compute; same considerations apply also to the *gMoE* with 29550 parameters compared to *SM* with 31660 parameters. Therefore there is no single model striking a much better trade-off, and more advanced approaches do not outperform simpler methods like *Ens*.

| setting | # experts | $\lvert\theta\rvert$ | $t_0$ | $t_1$ | $t_2$ | $t_3$ | $\lvert\theta\rvert$ | $t_0$ | $t_1$ | $t_2$ | $t_3$ |
|---------|-----------|------|-------|-------|-------|-------|------|-------|-------|-------|-------|
| | | | *Base* model perplexity | | | | | *Large* model perplexity | | | |
| *SM*_w1 | 4 | 65M | 28.57 | 27.45 | 26.91 | 26.53 | 210M | 22.47 | 21.62 | 20.84 | 20.54 |
| *SM*_w3 | 8 | 91M | * | * | 25.18 | | 323M | * | * | 19.29 | |
| *SM*_w4 | 12 | 116M | * | * | * | 24.41 | 436M | * | * | * | 19.01 |
| *Ens*_w1 | 4@2 | 130M | 26.20 | 25.12 | 24.57 | 24.35 | 420M | 20.32 | 19.55 | 19.14 | 18.92 |
| | 4@4 | 260M | 25.03 | 24.03 | 23.45 | 23.29 | 840M | 19.27 | 18.52 | 18.22 | 18.07 |
| *gEns*_w1 | 4@1 | 65M | 28.57 | | | | 210M | 22.47 | | | |
| | 4@2 | 130M | | 26.27 | | | 420M | | 20.25 | | |
| | 4@3 | 195M | | | 25.41 | | 630M | | | 19.49 | |
| | 4@4 | 260M | | | | 25.01 | 840M | | | | 19.18 |
| *gMoE*_w1 | 4 | 65M | 28.57 | | | | 210M | 22.47 | | | |
| | 6 | 78M | | 26.46 | | | 266M | | 21.22 | | |
| | 8 | 91M | | | 25.66 | | 323M | | | 20.39 | |
| | 12 | 116M | | | | 25.28 | 436M | | | | 20.15 |

Table 3: Large scale language modeling results. For *Ens* and *gEns*, 4@3 means 3 components in the ensemble, each of which has 4 experts per block, for instance.

The results on the large scale language modeling task reported in Tab. 3 show that bigger models perform better (the larger the number of parameters the lower the PPL for a given model class) and are also more statistically efficient (for instance the base *SM*_w1 attains 26.53 after seeing the whole stream, while the large *SM*_w1 obtains 22.47 just after seeing the first chunk of data), consistent with recent related work (Kaplan et al., 2020b; Li et al., 2020a). We also observe that *Ens* is a strong performer, with *Ens*_w1 and *gEns*_w1 models dominating *SM* models in all settings. Surprisingly, ensembles trained on distinct data chunks (*gEns*_w1; $t_1$ or $t_3$) perform no better than ensembles trained on a single data chunk (*Ens*_w1; $t_0$). For instance, among Base 2-model ensembles (4@2), *Ens*_w1 achieves a perplexity of 26.20 using a single data chunk ($t_0$), while *gEns*_w1 achieves a perplexity of 26.27 using models trained on each of the two data chunks ($t_1$). Finally, if test time inference is a concern, then *gMoE* is a preferable choice since its runtime is comparable to *SM*.

# 6 CONCLUSION AND PERSPECTIVES

In this work we introduced the anytime learning at macroscale (ALMA) setting, which is an instance of anytime learning under the assumption that data is observed as a sequence of large batches. ALMA better mimics the learning scenarios faced by machine learning practitioners, who want to efficiently solve a task, but time to time they receive more data to train on. We introduced metrics that enable the assessment in terms of error rate, memory usage and compute throughout the entire learning experience. Equipped with these tools, we then evaluated several approaches on three different datasets, including large scale language modeling. We found that methods that update parameters at an intermediate rate tend to yield a better trade-off, and that bigger models tend to generalize better. In particular, models that grow capacity over time generalize better particularly when the initial model is smaller, and ensembling is a very strong baseline.

A cynical interpretation of our finding that bigger models generalize better, could take the reader to the conclusion that it can all be solved by starting with a big model. However, as data is added over time so is computation. It is often the case that researchers working on large-scale learning instantiate the biggest possible model to train on their task, but few months later they can manage to launch even bigger models thanks to compute and engineering advances. How can the larger model leverage what has been learned from the previously trained model? Is there a modeling choice that strikes a better trade-off than retraining from scratch? More generally, what are good approaches to extract information from a new batch of data to integrate it into an existing model? While we do not provide a full answer to these questions, we do offer a framework to study them and several strong baseline approaches to compare against and build upon.

## 7 REPRODUCIBILITY STATEMENT

We have made several efforts to ensure that the results provided in the paper are fully reproducible. We first provide a clean codebase from which all the computer vision results in this paper are generated. In this codebase, one can find the exact hyperparameters used for each method in the provided configurations. We have attached a readme to the code in order to guide users running our code. For the LM experiments, as stated in the appendix we use the fairseq (Ott et al., 2019) and provide the required information to replicate our results.

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

APPENDIX

## A  GROWING MIXTURES OF EXPERTS

**Growing Mixture of Experts (*gMoE*):**  A mixture of expert (*MoE*) is a sequence of non-linear functions, each of which is potentially a mixture of experts (omitting the dependence on parameters):

$$m(x) = f^l(f^{l-1}(\dots f^1(x)\dots)), \text{ with } f^i(z) = \sum_{j=1}^{k} g^i(j|z)h^i(z|j)$$

where $g^i$ is the gating function at the $i$-th layer which outputs a categorical distribution over the number of experts, and $h^i(\cdot|j)$ is the $j$ expert at layer $i$. The gating function can be "soft" in which case it outputs non-zero weights for each expert via a softmax, or "hard" version in which case only one expert is selected through a multinomial sampling (and learned through the straight-through estimator in this paper (Bengio et al., 2013)). At test time in the "hard" case, we select the expert with the largest probability. The interest of mixtures of experts is they have a high expressivity, and experts can be easily added to increase the capacity of the model. The *gMoE* model is the growing version where, at each stage as illustrated in Fig. 3, new experts are added at each layer – details about the precise expansion process are given in Appendix.

The key design considerations are: *when* to grow, *what* to grow and *how* to grow. Here, we will refer to our default setting which favors simplicity, unless otherwise specified.

A growth step is triggered at each stage, ensuring a linear growth over time. We grow by adding one expert at each layer, making sure that all experts within a layer have the same architecture albeit with different parameters. In order to grow, we look at which expert has associated the largest cumulative loss; we call such expert the *losing* expert. The cumulative loss is defined as the sum of the losses of examples on the validation set that have been routed through a particular expert; each expert has associated a cumulative loss value. The rationale is to identify at each layer the expert responsible for the largest contribution to the total loss.

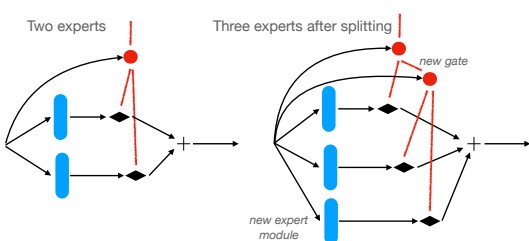

Figure 3: Illustration of a growth step in a tree structured mixture of experts. A network is composed of several layers like this. The blue squares are experts (e.g VGG layers). The red elements corresponds to the gatings which, given an input compute a score for each expert. When splitting an expert (right), the gating structure is updated by creating a child gate, and an additional expert is added to the mixture.

To avoid drop in the loss function and to keep its differentiability when splitting an expert, we propose a tree-based approach we the losing expert is split such expert into two experts with exactly the same parameters as illustrated in Fig. 3: Two children leaves are derived and we instantiate a new gating for the children which decides whether an input example routed to the old expert, should now go to the right or left expert child. The parameters of the new gate are initialized at random while the parameters of the new experts are exact copies of the ones of the losing expert that we split.

More formally, if $s$ is the losing expert then the term $g^i(s|z)h^i(z|s)$ is replaced by:

$$\sum_{k=1}^{2} g^i(s|z)g^i(k|z,s)h^i(z|s,k) \tag{3}$$

where $g^i(k|z,s)$ is the newly introduced gate, and $z$ is the input of the gating and experts.

Over time, the gating function learns to partition its input space into a binary tree (if we start from a single expert), and the gating value of an expert is the product of the gating probabilities on the path from root to the leaf expert. Both the gating tree structure and the particular initialization scheme guarantee that the growth step is smooth and fully differentiable, in particular, the loss before and after the growth step is the same.

If we consider each path in the MoE model to be a different model, then with $L$ layer of $k$ MoE components, there are $k^L$ many possible paths through the MoE model, hence the number of paths grows exponentially with the number of layers. You can think of this as an ensemble with exponentially many components, but this is still tractable because components share parameters.

---

**Algorithm 2** gMoE

---

 1: $k$: number of mega-batches to aggregate
 2: $\mathcal{D} = \emptyset$
 3: **function** TRAIN($\mathcal{D}_i$, $i$)
 4:     $\mathcal{D} \mathrel{+}= \mathcal{D}_i$
 5:     **if** $i$ mod $k == 0$ **then**
 6:         Extract $\mathcal{D}^{\text{VAL}}$ and $\mathcal{D}^{\text{TR}}$ from $\mathcal{D}$
 7:         **while** $m$ is not converged: **do**
 8:             $(x, y) \sim \mathcal{D}^{\text{TR}}$                      ▷ In practice, sample mini-batches.
 9:             m.update($x, y$)
10:         $\mathcal{D} = \emptyset$
11:         m.grow($\mathcal{D}^{\text{VAL}}$)                 ▷ Growth step can be done at a different rate too.
12: **function** GROW($\mathcal{D}^{\text{VAL}}$)
13:     **for** each layer in the network **do**
14:         Let $i$ be the losing expert on $\mathcal{D}^{\text{VAL}}$, i.e. the expert incurring the largest cumulative loss.
15:         Turn corresponding gating output in an internal node and derive 2 gate children
16:         Initialize the new experts by copying the parameters from the old parent expert.
17:         Initialize the new gating between the two siblings at random.

---

# B  HYPER-PARAMETER SETTINGS

## B.1  COMPUTER VISION EXPERIMENTS

For each megabatch received, we keep 10% of the data to perform cross-validation. All experiments are run on a single 16GB Quadro GP100 GPU. We apply data normalization for each dataset considered. A training minibatch size of 128 is used. *UMix* and *Ens* models have $N = 5$ in all experiments. for *gEns*, we train one model $n = 1$ at every mega-batch, so the total number of models depends on the amount of mega-batches. For Firefly we use a growth rate of $0.25$, meaning that at every growth phase, we add approximately a quarter of the initial number of parameters.

### B.1.1  MNIST

Models are trained for 100 epochs, and we report results with soft gating. We use the AdaDelta (Zeiler (2012)) optimizer with default learning rate of 1. We use a MLP with 2 hidden layers of varying width (e.g. 4,8 or 32 neurons).

### B.1.2  CIFAR-10

Models are trained for 200 epochs, as this was shown to be long enough to allow the model to converge with a learning rate of 0.01. We use Stochastic Gradient Descent with momentum value of 0.9 and weight decay of $1 \times 10^{-4}$. During training, we apply random horizontal flips and select random image crops with padding of 4 pixels. For the architecture, we use the same reduced VGG with batch normalization as prescribed in Wu et al. (2020). All layers are initialized with the same number of channels (e.g. 4, 8, or 32 channels). For the Firefly experiments, we keep all the Firefly-specific hyperparameters to the default values suggested in the author's public codebase. We make one exception to this, namely we adapt the growth ratio to result in linear (rather than exponential) growth.

## B.2  LANGUAGE MODELING EXPERIMENTS

All the language models are trained using fairseq (Ott et al., 2019) with a maximum of eight 32GB GPUs (NVIDIA V100), optimized with Adam (Kingma & Ba, 2014) using $\beta_1 = 0.9$, $\beta_2 = 0.98$,

$\epsilon =$ 1e-8. The learning rate is warmed up over the first several hundred updates (between 500 and 4000) and then linearly decayed to 0 over the remaining updates, with a peak value tuned between 2e-4 and 5e-3. Models are trained up to 120,000 updates with local batch size of 8 sequences per GPU, with gradient accumulation as needed to achieve a total batch size of 192 sequences; each sequence has 512 tokens. We fix the Switch Transformer balancing loss term to 0.01 and use a capacity factor of 1, following Fedus et al. (2021).

## C  ADDITIONAL COMPUTER VISION RESULTS

In this section we show the impact of several variants of our framework. Namely, we report results for (a) a varying number of mega-batches, (b) whether to use preemption or not, and (c) whether to initialize from scratch or simply finetuning when replay is performed.

### C.1  CIFAR

In the following results, we vary the number of megabatches. Below you can find results for $MB = 20$

#### C.1.1  DIFFERENT MBS

| Method | $|\theta|$ | Cum. Train TFLOPS | Inference MFLOPS | CER |
|---|---|---|---|---|
| SM | 2510 | 3.6 (20.0) | 0.4 | 2705.8±132.5 (2643.1±39.2) |
| | 11710 | 15.0 (82.5) | 1.7 | 2038.2±32.2 (1842.3±51.0) |
| | 31660 | 38.6 (212.5) | 4.3 | 1763.5±42.7 (1560.6±6.9) |
| | 140970 | 164.9 (907.0) | 18.3 | 1524.3±23.5 (1305.0±14.9) |
| | 1358510 | 1546.4 (8505.3) | 171.8 | 1307.3±19.1 (1118.4±17.0) |
| Ens | 12550 | 18.2 (99.8) | 10.1 | 2440.3±29.3 (2311.1±26.0) |
| | 704850 | 824.5 (4534.9) | 458.1 | 1230.2±9.5 (1045.9±9.8) |
| UMix | 12550 | 18.2 (99.8) | 10.1 | 2087.0±36.1 (1840.3±34.8) |
| | 704850 | 824.5 (4534.9) | 458.1 | 1502.3±36.0 (1286.4±27.7) |
| gEns | 2510 → 25100 | 3.6 (20.0) | 0.4 → 4.0 | 2726.9±70.3 (2542.1±24.1) |
| | 140970 → 1409700 | 164.9 (907.0) | 18.3 → 183.2 | 1348.3±16.8 (1282.9±8.5) |
| gMoE | 5214 → 29550 | 10.1 (65.3) | 0.5 → 1.7 | 2400.6±61.1 (2089.3±28.5) |
| | 275442 → 1485690 | 555.2 (3638.9) | 26.2 → 97.2 | 1448.1±29.3 (1222.4±17.5) |
| FF | 2446 → 32519 | 26.1 (222.3) | 0.4 → 5.5 | 2272.5±26.5 (1946.9±44.0) |
| | 140458 → 1646809 | 646.3 (7166.3) | 18.3 → 138.9 | 1449.7±35.7 (1189.3±19.6) |

Table 4: CIFAR-10 $MB = 10$ results in the paper

| Method | $|\theta|$ | Cum. Train TFLOPS | Inference MFLOPS | CER |
|---|---|---|---|---|
| SM | 2510 | 3.6 (38.1) | 0.4 | 2047.1±28.0 (1867.6±35.6) |
| | 140970 | 164.9 (1731.5) | 18.3 | 1171.2±52.9 (944.1±16.7) |
| UMix | 12550 | 18.2 (190.6) | 10.1 | 1615.1±32.1 (1300.2±25.0) |
| | 704850 | 824.5 (8657.5) | 458.1 | 1233.9±23.9 (876.4±29.8) |
| gEns | 2510 → 50200 | 3.6 (38.1) | 0.4 → 8.1 | 2000.5±30.8 (1833.9±25.2) |
| | 140970 → 2819400 | 164.9 (1731.5) | 18.3 → 366.5 | 1047.5±47.1 (949.6±48.7) |
| FF | 2446 → 22353 | 8.7 (315.4) | 0.4 → 1.4 | 1788.8±57.9 (1446.1±42.9) |
| | 140458 → 803761 | 300.9 (9646.7) | 18.3 → 47.7 | 1140.4±38.8 (799.2±19.8) |

Table 5: CIFAR-10 $MB = 20$ results

### C.1.2 PREEMPTED RESULTS

We also consider the use of a patience term when training the model. When the validation accuracy has not improved over 25 consecutive epochs, we stop training for the given learning phase. As expected, we observe gains on compute efficiency, with a small loss in performance.

| Method | $|\theta|$ | Cum. Train TFLOPS | Inference MFLOPS | CER |
|---|---|---|---|---|
| SM | 2510 | 1.2 (6.8) | 0.4 | 2844.6±36.5 (2788.2±76.9) |
| | 11710 | 4.4 (25.9) | 1.7 | 2223.0±41.9 (1929.3±43.3) |
| | 31660 | 11.2 (64.1) | 4.3 | 1898.6±33.6 (1615.8±23.5) |
| | 140970 | 39.9 (321.2) | 18.3 | 1574.6±36.3 (1374.9±25.8) |
| | 1358510 | 416.4 (2699.7) | 171.8 | 1366.1±28.6 (1158.1±17.5) |
| Ens | 12550 | 10.7 (61.9) | 10.1 | 2482.0±37.8 (2334.6±18.2) |
| | 704850 | 330.0 (3192.1) | 458.1 | 1238.7±13.2 (1048.5±8.0) |
| UMix | 12550 | 5.2 (35.5) | 10.1 | 2379.7±37.4 (1998.9±33.9) |
| | 704850 | 209.4 (1617.3) | 458.1 | 1675.9±57.8 (1376.4±51.5) |
| gEns | 2510 → 25100 | 1.0 (5.6) | 0.4 → 4.0 | 3048.6±171.0 (2688.4±36.0) |
| | 140970 → 1409700 | 42.8 (302.4) | 18.3 → 183.2 | 1432.8±33.7 (1325.2±19.9) |
| gMoE | 5214 → 29550 | 3.0 (25.6) | 0.5 → 1.7 | 2631.9±92.5 (2258.0±46.7) |
| | 275442 → 1485690 | 107.7 (1276.5) | 26.2 → 97.2 | 1562.7±38.0 (1298.4±30.9) |
| FF | 2446 → 35843 | 4.5 (100.3) | 0.4 → 3.1 | 2570.3±119.2 (2022.9±85.4) |
| | 140458 → 1622818 | 148.1 (2819.9) | 18.3 → 134.2 | 1537.9±36.5 (1253.3±43.0) |

Table 6: CIFAR-10 $MB = 10$ results, with preemption

| Method | $|\theta|$ | Cum. Train TFLOPS | Inference MFLOPS | CER |
|---|---|---|---|---|
| SM | 2510 | 0.8 (9.9) | 0.4 | 2404.0±71.1 (2002.3±28.0) |
| | 140970 | 32.9 (431.7) | 18.3 | 1242.7±43.2 (965.8±17.6) |
| Ens | 12550 | 9.3 (121.2) | 10.1 | 1828.3±20.4 (1646.8±14.5) |
| | 704850 | 257.1 (4788.2) | 458.1 | 962.1±20.6 (724.3±22.3) |
| UMix | 12550 | 4.1 (53.3) | 10.1 | 1820.3±55.2 (1377.1±51.3) |
| | 704850 | 197.8 (2459.3) | 458.1 | 1334.5±46.8 (931.4±21.2) |
| gEns | 2510 → 50200 | 0.9 (10.0) | 0.4 → 8.1 | 2325.4±93.3 (1944.2±60.8) |
| | 140970 → 2819400 | 34.5 (482.0) | 18.3 → 366.5 | 1133.5±29.7 (932.7±27.1) |
| FF | 2446 → 17359 | 2.0 (71.6) | 0.4 → 1.4 | 2034.3±44.0 (1576.5±97.9) |
| | 140458 → 791701 | 57.9 (2312.4) | 18.3 → 50.2 | 1249.7±65.1 (886.9±28.7) |

Table 7: CIFAR-10 $MB = 20$ results, with preemption

### C.1.3 INITIALIZING FROM SCRATCH

Below we show results, comparing the performance of re-training models from scratch on all the data seen so far vs simply finetuning the current model(s) on all the data. Main numbers are **finetuned models**, numbers in parentheses are **trained from scratch**.

Table 8: CIFAR-10 $MB = 10$ results with Replay. Numbers in () are models (re)initialized from scratch at the start of a new MB

## C.2 MNIST

| Method | $|\theta|$ | Cum. Train TFLOPS | Inference MFLOPS | CER |
|---|---|---|---|---|
| SM | 3210 | 0.02 (0.09) | 0.003 | $1054_{\pm25}$ ($1062_{\pm61}$) |
| | 16330 | 0.09 (0.48) | 0.016 | $631_{\pm10}$ ($616_{\pm9}$) |
| | 26506 | 0.14 (0.79) | 0.026 | $582_{\pm12}$ ($554_{\pm5}$) |
| | 132127 | 0.71 (3.92) | 0.132 | $504_{\pm8}$ ($485_{\pm4}$) |
| | 269322 | 1.45 (7.98) | 0.269 | $497_{\pm8}$ ($474_{\pm4}$) |
| Ens | 16050 | 0.09 (0.47) | 0.080 | $870_{\pm9}$ ($885_{\pm18}$) |
| | 132530 | 0.71 (3.93) | 0.661 | $517_{\pm6}$ ($493_{\pm5}$) |
| UMix | 16050 | 0.09 (0.47) | 0.080 | $668_{\pm17}$ ($646_{\pm18}$) |
| | 132530 | 0.71 (3.93) | 0.661 | $508_{\pm10}$ ($486_{\pm5}$) |
| gEns | $3210 \rightarrow 32100$ | 0.02 (0.09) | $0.003 \rightarrow 0.032$ | $1020_{\pm39}$ ($1027_{\pm29}$) |
| | $26506 \rightarrow 265060$ | 0.14 (0.79) | $0.026 \rightarrow 0.264$ | $551_{\pm6}$ ($543_{\pm7}$) |
| gMoE | $7950 \rightarrow 50610$ | 0.16 (1.08) | $0.008 \rightarrow 0.050$ | $863_{\pm17}$ ($766_{\pm16}$) |
| | $54318 \rightarrow 304626$ | 0.97 (6.55) | $0.054 \rightarrow 0.304$ | $573_{\pm8}$ ($535_{\pm5}$) |

Table 9: MNIST Results from the paper, with $MB = 10$

| Method | $|\theta|$ | Cum. Train TFLOPS | Inference MFLOPS | CER |
|---|---|---|---|---|
| SM | 16330 | 0.04 (0.19) | 0.016 | $637_{\pm10}$ ($608_{\pm10}$) |
| | 132127 | 0.25 (1.84) | 0.132 | $507_{\pm5}$ ($490_{\pm5}$) |
| | 269322 | 0.50 (3.11) | 0.269 | $491_{\pm6}$ ($478_{\pm5}$) |
| Ens | 16050 | 0.06 (0.44) | 0.080 | $880_{\pm14}$ ($877_{\pm20}$) |
| | 132530 | 0.37 (3.05) | 0.661 | $521_{\pm4}$ ($496_{\pm5}$) |
| UMix | 16050 | 0.04 (0.19) | 0.080 | $687_{\pm18}$ ($636_{\pm21}$) |
| | 132530 | 0.25 (1.85) | 0.661 | $507_{\pm7}$ ($488_{\pm4}$) |
| gMoE | $7950 \rightarrow 50610$ | 0.06 (0.44) | $0.008 \rightarrow 0.050$ | $849_{\pm28}$ ($829_{\pm38}$) |
| | $54318 \rightarrow 304626$ | 0.33 (2.71) | $0.054 \rightarrow 0.304$ | $572_{\pm11}$ ($534_{\pm6}$) |

Table 10: MNIST $MB = 10$ using preemption with patience value of 25 epochs

| Method | $|\theta|$ | Cum. Train TFLOPS | Inference MFLOPS | CER |
|---|---|---|---|---|
| SM | 16330 | 0.48 (0.48) | 0.016 | $615.6_{\pm9.0}$ ($591.0_{\pm8.7}$) |
| | 132127 | 3.92 (3.92) | 0.132 | $485.5_{\pm3.6}$ ($481.2_{\pm3.3}$) |
| | 269322 | 7.98 (7.98) | 0.269 | $474.0_{\pm3.7}$ ($473.3_{\pm5.8}$) |
| Ens | 16050 | 0.47 (0.47) | 0.080 | $885.4_{\pm18.4}$ ($881.6_{\pm23.3}$) |
| | 132530 | 3.93 (3.93) | 0.661 | $493.1_{\pm4.9}$ ($495.6_{\pm2.3}$) |
| UMix | 16050 | 0.47 (0.47) | 0.080 | $645.9_{\pm18.1}$ ($617.1_{\pm14.6}$) |
| | 132530 | 3.93 (3.93) | 0.661 | $485.6_{\pm5.0}$ ($480.6_{\pm4.2}$) |
| gMoE | $7950 \rightarrow 50610$ | 1.08 (1.08) | $0.008 \rightarrow 0.050$ | $765.6_{\pm15.9}$ ($778.6_{\pm16.0}$) |
| | $54318 \rightarrow 304626$ | 6.55 (6.55) | $0.054 \rightarrow 0.304$ | $535.4_{\pm5.2}$ ($543.0_{\pm7.7}$) |

Table 11: MNIST $MB = 10$ results with replay, numbers in parentheses are models initialized from scratch at the start of every MB

