# OpenReview forum: "On Anytime Learning at Macroscale"
_ICLR.cc/2022/Conference — ICLR 2022 Submitted_

### Official Review · Reviewer_x4p7 · 2021-10-24

**Correctness:** 3
**Technical Novelty And Significance:** 2
**Empirical Novelty And Significance:** 2
**Recommendation:** 5
**Confidence:** 4

**Main Review:**

The problem setting of anytime learning at macroscale is interesting and novel to me. How to efficiently learn data in a streaming fashion is a practical challenge. The proposed learning setting targets the level of the entire sequence of large datasets.

Although the method overall is valuable and interesting, the paper is poorly organized and thus hard to understand. It is difficult to find some critical details or notation definitions that are related but separated apart in the paper. The method lacks an integrated and principal formulation from which the techniques are derived. Though they show some theoretical results, it is hard to relate them to the objective and the algorithm steps explicitly and tightly. The contribution is unclear. Here are some detailed comments:

- What does "organically generated" mean?
- " both training ... and finetuning ... are not satisfying" should use "neither...nor..."
- What does "constrained capacity" ?
- What is the definition of macroscale?
- Unclear the difference between ALMA and other learning frameworks.
- Related work compares ALMA to lots of different prior work, but it is poorly organized and I am not sure why those prior work should be considered as comparisons.
- It is suspicious to say that "extensions to regression and unsupervised learning (where y is missing) are trivial".
- In fixed architecture, why "A potential drawback of Ens is that evaluation and training are inconsistent"?
- In Growing Mixture of Experts, "Compared to Ens, MoE has exponentially many more components". I am not sure where "exponentially" comes from.


Quality: The submission is technically sound. The claims in the contribution are supported by empirically results. It is a complete piece of work.

Clarity: The experimental details are also very specific, such that reproducing the results should be possible.

**Summary Of The Paper:**

Summary: This paper proposes anytime learning at macroscale (ALMA), which is anytime learning under the assumption that data is observed as a sequence of large batches. This paper introduces metrics that can be used to access the error rate, memory, and compute throughput the entire learning process. They evaluate multiple learning models on different datasets in the ALMA setting. They observe that methods that update parameters at a moderate rate tend to yield a better tradeoff, while bigger models tend to generalize better.

**Summary Of The Review:**

This paper proposes a novel learning setting called anytime learning at macroscale (ALMA). The proposed idea is simple and technically sound. Extensive evaluations are conducted. However, the contribution is unclear, and the presentation needs improvement.

---

> ### Author Response · Authors · 2021-11-19
> **Response to Reviewer x4p7**
>
> We thank you for your review. Please see the general comments.
>
> *Unclear contributions*
>
> Please see the last three paragraphs of section 1. In short, we introduce a new learning setting (ALMA) which is distinct from online learning and continual learning (see response to reviewer D3ZG as well as Figure 1) and we report an extensive empirical evaluation of several baseline approaches, including both fixed and dynamic architectures while varying model size, amount of replay, waiting time, and scale of the problem.
>
> *Unclear presentation*
>
> We are very sorry to hear this, as we actually spent a lot of effort in polishing the writing. Could you please give us an example for “It is difficult to find some critical details or notation definitions”? And when saying “The method lacks an integrated and principal formulation”, which method are you referring to? We had trouble revising the paper as we are not sure what was unclear.
>
> Now, responding to your detailed comments (these answers have been integrated in the revised version of the paper):
>
> - `organically generated`
> We meant content produced by people on the internet as part of their daily interaction, as opposed to content that people were tasked to do.
>
> - `What does "constrained capacity" mean?`
> It means that a small model with a fixed architecture (constant capacity) might end up underfitting as more and more data is observed. This contrasts with a model that has a dynamic architecture. (see correction in the paper)
>
> - `Definition of macro-scale`
> We used the term macro-scale to indicate the level of granularity of our analysis, which is dictated by the arrival time of mega-batches as opposed to individual data points (or mini-batches). In classical anytime learning, we would look at the anytime performance while learning from each mega-batch (if I stop training after N mini-batches, what error rate do I get?). Instead, we focus on the anytime performance at the stream level where ticks are given by mega-batches. By doing so, we assume negligible the time to learn from each mega-batch and only report the learning curve across mega-batches (If I stop training after M mega-batches, what is the error rate?), folding the cost of learning on each mega-batch in the computational cost.
>
> - `Related work compares ALMA to lots of different prior work, but it is poorly organized and I am not sure why those prior work should be considered as comparisons.`
> Please see answers above to R1 and R2 on the fundamental difference with online learning, update in the paper and added diagram.
>
> -`It is suspicious to say that "extensions to regression and unsupervised learning (where y is missing) are trivial".`
> We can say this because our framework is *agnostic* to the objective used to train the model. The framework is only about how data is given to the learner (in large chunks) and how performance should be measured (anytime performance at the level of chunks taking into account memory and compute). For example, the language modeling experiments are fully unsupervised, yet the same analysis in ALMA can be done.
>
> - `why "A potential drawback of Ens is that evaluation and training are inconsistent"?`
> It is inconsistent because the scoring at test time is done via averaging, while at training time each model is trained independently (there is no back-propagation through the average score).
>
> - `In Growing Mixture of Experts, "Compared to Ens, MoE has exponentially many more components". I am not sure where "exponentially" comes from.`
> If we consider each “path” in the MoE model to be a different model, then with `L` layer of `k` MoE components, there are `k^L` many possible paths through the MoE model, hence the number of paths grows exponentially with the number of layers. You can think of this as an ensemble with exponentially many components, but this is still tractable because components share parameters. We have clarified this in Appendix A.

---

### Official Review · Reviewer_KMi2 · 2021-10-25

**Correctness:** 4
**Technical Novelty And Significance:** 2
**Empirical Novelty And Significance:** 3
**Recommendation:** 6
**Confidence:** 3

**Main Review:**

The problem presented by the authors is relevant to applied ML/AI problems, which are always a work in progress. Further, improving the efficiency of learning is desirable. So, the problem is reasonably well motivated.

1. I'm skeptical about the value of the cumulative error rate. From a practical point of view a data engineer might be concerned with the questions; how good is the model I have now? What would be impact of further data collection? If one has collected a set of batches, the performance of prior models is not terribly relevant.

2. The non-iid nature of the data is mentioned, and it is mentioned that cross-validation is carried out only on the current batch. These concepts could be explored more thoroughly. What are the issues with evaluation in this setting? Should I hold out a portion of every batch to use for evaluation? What other evaluation strategies are possible? What are their strengths and weaknesses?

2. In continuous streaming settings, there may be distribution shift over time. That is not addressed in this work.

3. The authors make a big point about the scale of the problem. Obviously this makes naive approaches less appealing. But how does the problem scale? What really separates (if anything), the macroscale problem from more mundane sized problems? Are there underlying scaling laws at work that cause shifts the performance of each learning strategy?

**Summary Of The Paper:**

The authors consider a batch learning problem, in which large batches of data arrive in series. They explore the performance of several types of algorithms in terms of their computational cost, model size, and error rate.

**Summary Of The Review:**

The paper addresses a real need, but more insight is needed.

---

> ### Author Response · Authors · 2021-11-19
> **Response to Reviewer KMi2**
>
> We thank you for your review.
>
> *On the Cumulative Error Rate (CER)*
>
> This metric is used to determine the overall performance of the system throughout its lifespan. As you pointed out, at each step what matters is the performance on the current data. If we want to compare two systems throughout their learning experience then we simply sum their performances over time. In other words, if two systems obtain the same error rate by the end of the learning experience, we would prefer the one that has used the least resources or has converged faster (and to know that we need to look at how well they did in the past) -- see update in Section 3.1
>
> *On cross-validation in ALMA*
>
> Thank you for the suggestion. The questions you raised are interesting; in this paper we tried two strategies. We either a) cross-validated only on held-out data from the current megabatch, or b) on the concatenation of all held-out data from the current and previous megabatches (e.g. when using replay). We found that while both approaches give reasonable performance, storing data from previous mega-batches (for both cross-validation and training) works best (see Tab. 2 : replay vs non-replay results).
> On distribution shifts across mega-batches
> It is true that in some settings, a change in distribution can be encountered over time. In this work, we choose to first focus on the case where no distribution shifts are observed, which is simpler and still quite practical. As we can see from our empirical evaluation, there are still important challenges (e.g. when and how to retrain a model, how to grow its capacity), requiring more investigation despite the  i.i.d setting. Moreover, as stated in the updated introduction, it corresponds to a real setting encountered in production pipelines.
>
> *On the macroscale aspect of ALMA*
>
> We used the term macro-scale to indicate the level of granularity of our analysis, which is dictated by the arrival time of mega-batches as opposed to individual data points. In classical anytime learning, we would look at the anytime performance while learning from each mega-batch (if I stop training after N mini-batches, what error rate do I get?). Instead, we focus on the anytime performance at the stream level where ticks are given by mega-batches. By doing so, we assume negligible the time to learn from each mega-batch and only report the learning curve across mega-batches (If I stop training after M mega-batches, what is the error rate?), folding the cost of learning on each mega-batch in the computational cost. (see updated introduction)
>
> Thank you.

---

> > ### Comment · Reviewer_KMi2 · 2021-11-24
> > **Still wondering about mega vs mini**
> >
> > I have read the other reviews and author responses. Generally, the author responses are satisfying. However, I'm still wondering about the differences between macro- and non-macro-scale. What I'd really like to know is whether any fundamental changes occur as one scales the size of a mega-batch up or down. If changes occurred, that would indicate some underlying scaling law and justify macro-scale as it's own problem. As the authors point out in their response, we have the classical setting:
> > "if I stop training after N mini-batches, what error rate do I get?"
> > and the mega-scale setting
> > "If I stop training after M mega-batches, what is the error rate?"
> > The only thing separating these questions is mini vs mega. What I'm asking is: Is there any reason a single approach won't work well on both problems? The authors suggest that in the mega setting time to learn becomes negligible. I supposed that depends on how frequently the batches are coming in though.

---

> > > ### Author Response · Authors · 2021-11-24
> > > **Author Reply**
> > >
> > > Thanks for taking the time to read all of this and for following up. We truly appreciate it.
> > >
> > > 1) ALMA is not about learning with SGD using large mini-batches. The (SGD) mini-batch size is always the same in all experiments we ran. The critical difference between our setting and SGD training on a given dataset is that, in our case,  large chunks of data (i.e. what we call mega-batches) arrive in sequence and the learner does not have access to future mega-batches.
> > >
> > > 2) However, the learner can improve its performance by performing several passes over each mega-batch, because this is big relative to the size of the model and the amount of compute given. As mentioned in our previous response, on CIFAR 10 we can achieve 80% error rate by doing several passes as opposed to merely 53% if we were to do just one pass over each mega-batch. However, once the learner does multiple passes over each mega-batch then the data stream is not iid anymore.
> > >
> > > 3) Because of the non-stationarity induced by looping over each mega-batch multiple times and because the learner cannot get access to future data, there will be a gap in error rate by the end of the learning experience (w.r.t a classical training over all data at once).
> > >
> > > 4) Of course, if all we care about is the final error rate then retraining from scratch whenever a new chunk of data arrives is the upper bound performance. In this case, receiving data as a sequence of large chunks does not bring anything.
> > >
> > > 5) However, once we not only measure error rate (or even better, the area under the error rate curve which is useful to assess how quickly the model learns) but also compute, then it is not obvious anymore what method strikes a better trade-off: is it better to train on every new chunk of data [our experiments show that an intermediate rate works better]? Is it better to use replay [our experiments show that for a 10% reduction of error rate the computational cost grows by a factor of even 10]? Is it better to use a small model to start with [ideally, but current methods are not good enough]? what are efficient methods to train larger models [our experiments show that ensembling is often the best option]?
> > >
> > > 6) in practice like in the LM application we report, we do not have a choice on how to receive data but data just comes in chunks. The task is then to design a learning algorithm that is efficient and that can leverage what was learned on the previous chunks.
> > >
> > > - “Is there any reason a single approach won't work well on both problems?” Yes. Intuitively, the approach should build on the top of the previously learned (and fully converged) model, possibly expanding capacity over time. However, figuring out how to make this work is avenue of future work. This paper proposes the learning framework, metrics and baselines. The current paradigm of retraining from scratch bigger and bigger models over time is clearly great in terms of error rate but terrible in terms of compute.
> > >
> > >
> > > Please let us know if this answered your questions. Thanks again.

---

### Official Review · Reviewer_4ezA · 2021-11-02

**Correctness:** 3
**Technical Novelty And Significance:** 1
**Empirical Novelty And Significance:** 2
**Recommendation:** 5
**Confidence:** 4

**Main Review:**

Paper is well written. The authors document the approach they have considered, and the metrics used to evaluate the various experimental settings used.

As I started to read the paper, the problem setting seemed very similar to the ones used in data stream mining research over the past decade. Though the authors state that the primary differentiator to the stream setting is that the models use what they call as "meta-batches" as streams rather than streaming single data instances, I fail to understand any theoretical or empirical difference in the two approaches. There exists multiple popular data stream frameworks such as MOA (Massive Online Analysis) that is used exactly for the problem setting described in the paper. So, the primary contribution of the paper seems to be in extensively evaluating the model complexity and approaches over various data and problem settings. Moreover, majority of the future questions that the authors hope to answer have been studied across various papers (in similar forms). Please refer to Lu, Jie, et al. "Learning under concept drift: A review." IEEE Transactions on Knowledge and Data Engineering 31.12 (2018): 2346-236.

The second objection of the paper is that the authors seem to use simple datasets from today's standard to derive their conclusion. Though the conclusions in the paper are fair and not surprising, a more complex set of datasets may provide a stronger result. Furthermore, it is important to note that there are other factors that influence the classifier performance, more than the batch size and data size available for training. The data itself may be imbalanced, non standard etc. So, by using more datasets, these issues can potentially be elevated.

**Summary Of The Paper:**

The authors describe a framework to perform empirical evaluation of an anytime learning setting where data is available in a streaming minibatch fashion. With a primary aim to measure performance of a classifier across variety of practical settings of such streaming data to not only achieve high accuracy, but also provide non-trivial prediction anytime using limited computational resources. Using multiple benchmark datasets, the paper concludes that methods with intermediate parameter updates are better on the accuracy to computational efficiency tradeoff, and larger models generalize better.

**Summary Of The Review:**

In summary, the original contribution is not very clear. The authors have ignored to discuss comparisons with a branch of data stream mining reseach that has provided similar conclusion. And the empirical evaluation is on simple datasets, and may need further evidential support from more complex datasets.

---

> ### Author Response · Authors · 2021-11-19
> **Response to Reviewer 4ezA**
>
> We thank you for your review and for providing us with relevant literature on learning from streaming data. Please see the general comments regarding the positioning of ALMA with respect to other frameworks.
>
> *"1. There exists multiple popular data stream frameworks such as MOA (Massive Online Analysis) that is used exactly for the problem setting described in the paper."*
>
> MOA is a software package : it contains several implementations of methods designed for ML / data mining on evolving data streams. Note that the implemented methods are currently targeting mostly tabular data using shallow learning algorithms like  Naive Bayes and Decision Trees, without support for deep neural networks.
>
> The streaming learning setting of MOA is similar to ALMA, with the major difference that in MOA each example is seen once and only once while in ALMA the learner has the ability to revisit the same example multiple times (only access to future data is prohibited). In ALMA data comes in large chunks which are usually replayed several times.
>
>
> *"2. The open questions posed by this paper have been mostly answered in "Learning under concept drift: A review."*
>
> In our paper we have posed many open-ended questions. However, none are related to concept drift (i.e. a change in distribution). We are solely focused on how to combine knowledge over multiple datasets sampled from the same distribution.
>
> The first part of the referenced paper is about drift detection, the second half is about how to adapt to this drift once it’s detected. Instead, our work assumes that the distribution of each chunk of data is the same. The learner experiences non-stationarity only because it may decide to perform several passes over each chunk of data before processing the next one. There is no concept drift whatsoever in our work.
> We have added a reference and contrasted this work in the revision.
>
> *"3. The authors seem to use simple datasets from today's standard to derive their conclusion."*
>
> All the references linked by the reviewer use tabular data . The LM experiments use “real” data. The overall dataset has 1.7B tokens with a vocabulary of size 50,000. This is more than 3 orders of magnitude of what reported in the referenced paper! As stated in section 5, just the LM experiments of table 3 took over 100 GPU days to complete.
>
> Even the smaller scale experiments on MNIST and CIFAR10 are interesting, because they exhibit similar trends as the large-scale LM experiments enabling practitioners to test ideas quickly.
> The language modeling experiment does use imbalanced classes as the prior distribution over tokens is heavy-tailed.
>
> *4. Conclusions are not surprising.*
>
> We respectfully disagree. It may be intuitive that using an intermediate waiting time strikes the best trade-off, but how much is lost when deviating from this? And how does this change as we scale up the architecture size? Moreover, it may be intuitive that growing an architecture over time helps, but how does this compare to other approaches like ensembling? Does an ensemble whose components are trained on different chunks of data work better than an ensemble where components are trained on a different shuffling of the same chunk of data? Which approach best leverages an intermediate model trained on less data? If all of this is really not surprising, could you please provide a reference where they studied these questions?
>
> We hope the reviewer will revise their assessment after these clarifications. We are of course happy to further follow up.
>
> *Comparisons of ALMA with other frameworks :*
>
> Please see our answer to reviewer D3ZG.
>
> Thank you.

---

> > ### Comment · Reviewer_4ezA · 2021-11-25
> > **On author response**
> >
> > Thank you for your response. After reading all the responses, including those to other reviewers, I agree with the answers provided on certain questions on dataset size and differences with existing stream mining work. Though the problem addressed is different from concept drift and single stream mining methodologies, the concept of data arrival speed and data chunks considered for learning still seem similar. The paper at least needs a reasonable discussion on the differences/similarities and how such concepts can be applied here. This seems particularly important as the paper is positioned as a "Anytime Learning" scenario as per the title. I am still not convinced that there is a conceptual difference with continual/stream learning frameworks (even if ALMA has the ability to hold and use historically observed data). After reading though the objections raised by other reviewers and the corresponding author reponses, I have changed the rating accordingly.

---

### Official Review · Reviewer_D3ZG · 2021-11-03

**Correctness:** 1
**Technical Novelty And Significance:** 2
**Empirical Novelty And Significance:** 2
**Recommendation:** 3
**Confidence:** 4

**Main Review:**

My main concern is with the motivation of the setup. In particular, it does not seem to be very different from online learning. The paper does discuss the differences between the two setup, however the arguments are a bit superficial. In particular, it is claimed that in online learning the examples are streamed one at the time as opposed to receiving them in large batches - but this does not seem to make the online learning methods completely handicapped. Randomized algorithms (such as Thompson-sampling based methods) should perform reasonably. (For each example in batch i, apply the model obtained after processing all the examples in batches 1,2,..., i-1.) It would be great if the authors could elaborate on this.

**Summary Of The Paper:**

The paper introduces a novel setup called Anytime Learning at Macroscale. In this setup the learner receives the examples as a sequence of large batches, and is required to output a model after processing each batch. This model is used to give prediction for the next batch. The overall performance is then sum of the average losses on the individual batches.

**Summary Of The Review:**

Concern with the motivation: the setup is not that different from online learning after all.

---

> ### Author Response · Authors · 2021-11-19
> **Response to Reviewer D3ZG**
>
> We thank you for your review. Please see the general comments detailing the paper edits.
>
> *On differences with online learning:*
>
> In the paper (intro 2nd paragraph), we mention that in ALMA, the training data comes at a slower rate than the speed at which the model can process it. This is different from online learning, where it is assumed that data speed matches the processing speed. We have added a new figure (Fig. 1) to better illustrate this.
>
> This difference has significant consequences. In online learning, each example is processed once and only once by the model. Therefore, the stream of training data in the online setting is i.i.d, but this is not the case for ALMA. In the latter, the learner might perform multiple passes over each chunk of data in order to decrease the error rate as much as possible, although this makes the stream of data points not i.i.d..
>
> It follows that during training, models in the ALMA setting must decide when to further train a model, or when to wait for more data. Through our extensive empirical evaluation we found that this question (the frequency at which to retrain a model) to be the most significant parameter to strike a good anytime accuracy vs. compute tradeoff (see Figure 1). We note once again that this is not the case in the regular online learning setting.
>
> The two settings also differ in terms of the training/evaluation protocols. In online learning, a model first makes a prediction on the incoming data to determine its error, and then it updates its internal parameters (prequential approach). In ALMA instead, we extract a validation set from each chunk of data for tuning hyper-parameters (such as how many epochs over each chunk need to be performed), and then we compare models on the static hidden test set once the stream has been exhausted using model checkpoints saved throughout the learning experience.
>
> *Using online learning methods in ALMA:*
>
> If we simply transpose an online learning approach to the ALMA setting, such as online SGD where each datapoint is processed only once by the model, we avoid dealing with concerns on when to further train a model. However, doing so limits the final performance of the model. The online solution is equivalent to performing a single training epoch on the dataset (akin to using a very small waiting time), giving a 53% final test accuracy on CIFAR10 vs over 80% for the finetuning solutions developed for the ALMA setting.
>
> In general, ALMA is designed to encapsulate realistic constraints that emerge when training deep neural networks in settings where data is made available in stages. The language modeling application we report in the paper is one important example of this. The major practical questions we are addressing are:
> 1. Given some new data, is it better to wait for more data before further training? What are the trade-offs?
> 2. When training a large-scale model, is there a way to leverage a slightly smaller scale model trained on less data?
>
> In this work we make clear what the trade-off is for 1), and test several non-trivial baselines for 2). We have also released our code to let other people investigate the same framework.
>
> Please, let us know if we need to further clarify. We respect your judgement, but believe we have made a solid empirical contribution that should not deserve a rejection.
>
> *On Randomized algorithms / Thompson Sampling :*
>
> Thompson Sampling is used for online settings with bandits, with the intent to address the exploration-exploitation trade-off [1]. In our setting, the agent does not need to do any exploration, as it has no control over the data received.
> Regarding the approach you suggested, where the model trains on examples in batches 1,2,..., i-1, this is exactly the fine-tuning method described in the paper.
> Regarding Randomized algorithms, could you please elaborate on this or provide us with a reference ?
>
> [1] An Empirical Evaluation of Thompson Sampling, Chapelle & Li.
>
> Thank you.

---

> > ### Comment · Reviewer_D3ZG · 2021-11-27
> > **Some gaps are filled, but details seem to be missing**
> >
> > Thanks for the response, it did fill some gaps in my understanding of the paper. What is written about the differences with online learning and about using online learning methods is indeed interesting and important; in fact, it should also be discussed in the paper in depth as one of the main topics. (This goes for the experiments too.) Right now, however, I don't see such a discussion, just some mentions of some of the aspects here and there; contrary to what is written in the response "In this work we make clear what the trade-off is for 1), and test several non-trivial baselines for 2)". One clean way to address these questions in the paper could be to devise experiments for them explicitly to provide answers in a clean and measurable fashion.

---

### Author Response · Authors · 2021-11-19
**General Comments**

We thank the reviewers for their comments on the paper. We have addressed each reviewer individually, here we highlight key points and changes in the paper.

We have clarified the key differences between ALMA and online learning. We have added a new figure (Fig. 1) indicating how ALMA positions itself relative to other learning frameworks. We have also edited the text to better reflect these distinctions.

---

### Decision · Program_Chairs · 2022-01-20

**Decision:**

Reject

**Comment:**

The work presented in this submission is focused on a new approach for learning a model that can perform well at any point in time, and called Anytime Learning at Macroscale (ALMA). The algorithm processes data through a series of training batches, each of these processing steps being followed to a model evaluation. The total loss is the average (or sum) of the losses computed at each step.

Reviewers agreed that the paper is not ready for acceptance at ICLR 2022 as the presentation of the work lacks of clarity, especially w.r.t. to the similarities with online learning and the learning of streams of data, and the fundamental difference between small or moderate batch sizes and very large batches.